# Biodegradation of Methylene Blue Using a Novel Lignin Peroxidase Enzyme Producing Bacteria, Named *Bacillus* sp. React3, as a Promising Candidate for Dye-Contaminated Wastewater Treatment

**Van Hong Thi Pham** [1] **, Jaisoo Kim** [2]**, Soonwoong Chang** [3,*] **and Woojin Chung** [3,*]

[1] Department of Environmental Energy Engineering, Graduate School of Kyonggi University, Suwon 16227, Korea; vanhtpham@gmail.com

[2] Department of Life Science, College of Natural Science of Kyonggi University, Suwon 16227, Korea; jkimtamu@kyonggi.ac.kr

[3] Department of Environmental Energy Engineering, College of Creative Engineering of Kyonggi University, Suwon 16227, Korea

* Correspondence: swchang@kyonggi.ac.kr (S.C.); cine23@kyonggi.ac.kr (W.C.); Tel.: +82-31-249-9755 (W.C.)

**Abstract:** The emission of methylene blue (MB) from common industries causes risks to human health by making clean drinking water unavailable and hampering environmental safety. A biological approach offering a more cost-efficient and sustainable alternative solution has been studied and demonstrated to be significantly effective for the removal of MB using promising microbial isolates. Therefore, this study targeted bacterial candidates, namely *Bacillus* sp. React3, isolated from soil with the potential to decolorize MB. The phenogenic identification of strain React3 was performed by 16S rRNA sequencing, showing a similarity of 98.86% to *Bacillus velezensis* CR-502T. The ability of this bacterial strain to decolorize MB was proven through both the lignin peroxidase efficiency and accumulation in the biomass of the living cells. MB removal was determined by the reduction in the maximum absorption at a wavelength of 665 nm, which was observed to be up to 99.5% after 48 h of incubation. The optimal conditions for the MB degradation of strain React3 were pH 7, 35 °C, static, 4% inoculum, and 1000 mg/L of MB, with tryptone as a carbon source and yeast extract as a nitrogen source.

**Keywords:** biodegradation; MB; lignin peroxidase enzymes; *Bacillus* sp.; wastewater

## 1. Introduction

Dyes are used in different industries, such as textiles, food, rubber, printing, cosmetics, medicine, plastic, concrete, and the paper industry, for multiple purposes. These are considered major sources of water pollution. Among these, methylene blue (MB) dye (3,7-(dimethylamino)-phenothiazin-5-iumchloride) is widely used. Its chemical structure is described as an aromatic system conjugated to one or more (–N=N–) groups and ($SO_3$) groups associated with hydroxyl, methyl, chloro, triazine amine, and nitro groups [1,2]. MB is a recalcitrant, heterocyclic aromatic dye that is also used in microbial laboratories for inhibiting the growth of fungi and as a staining reagent for microbial analysis [3]. Nowadays, it is employed in dyeing cotton, nylon, rayon, and hair, and for the coloring of plastics, oils, and gasoline [4,5]. It is reported that 10–70% of the amount used is discharged into the aquatic ecosystem because of the inefficient dyeing processes [6,7]. The presence of MB in the range of 10–50 mg/L is considered to have devastating effects on the environment by reducing the fertility of soil and crop quality and decreasing the photosynthetic capacities of aquatic plants, leading to a loss of biodiversity and harming human health [8].

Based on the characteristics of MB, physiochemical methods such as flocculation, sedimentation, and adsorption have been used for the decolorization and detoxification of MB from wastewater, but they are insufficient for completely removing the dyes [9]. Additionally, at the same time of the treatment process, hazardous by-products with high levels of toxicity, such as waste, are also generated. Moreover, the treatment is expensive and requires energy. Thus, the use of biomaterials for bioremediation, such as green technologies, has intensified in recent years. Compared with physicochemical methods, biological approaches have numerous noteworthy advantages owing to their low cost, ease of operation, environmental friendliness, release of nontoxic metabolites, and lower water and energy requirements [10,11]. Bacteria have emerged as effective candidates for the processes of dye decolorization by breaking down organic pollutants and utilizing them as carbon and energy sources for their growth. In particular, many bacterial strains are highly tolerant to extreme toxic pollution conditions and have a short growth cycle [12,13].

Different microbial biomasses (live or dead) under various conditions (aeration, temperature, pH, salinity, and pressure) illustrate an important contribution in this regard [14]. Such microbial groups show the ability to degrade MB and belong to several genera, including *Bacillus, Pseudomonas, Staphylococcus, Acinetobacter, Galactomyces, Raoultella*, and *Aspergillus* [3,15–19]. Most studies have focused on the bacterial degradation of strains that grow fast, are widespread, and are highly tolerant. Kishor et al. [3] investigated an effective degrader to remove MB, *Bacillus albus*, isolated from textile sludge [3]. In another study, *Acinetobacter pittii* exhibited the ability to remove 73% of MB within five days [18]. Strain *Pseudomonas aeruginosa* demonstrated the degradation of MB from 82.25% to 97.82% with an increase in initial MB concentration from 50 to 200 mg/L over 24 h [17]. Isolates identified as *Comamonas aquatic* PMB-1 and *Ralstonia mannitolilytica* PMB-2 were able to decolorize MB at percentages of 67.9% and 60.3%, respectively, over 96 h at 37 °C [20]. In a previous study, the rate of removal of MB from wastewater by a microbial community of anaerobic granular sludge in the presence of predominantly *Candidatus Cloacimonetes* was over 90% [21].

Therefore, in recent years, the use of a microbial culture has been reported to achieve efficient dye degradation owing to its synergistic metabolic pathways. Additionally, enzymes such as lignin peroxidase secreted from bacteria are considered as biocatalysts that are suitable for dye degradation and color removal through the oxidation process. The isolation sources of bacteria are generally from the places in which the functional bacteria play a role in MB-contaminated wastewater or polluted soil. However, the normal soil, especially the soil from the surrounding plant roots, is still a material pool for research, possessing 99% of unculturable bacteria that have not yet been discovered [22]. Thus, this study aims to investigate the potential MB-decolorizing bacterial candidates from soil in an artificial MB-contaminated aqueous solution at a laboratory scale. Strong degrader *Bacillus* sp. React3 was chosen in the subsequent steps for the analysis. Thus, an environmentally friendly and safe alternative method for bioremediation by both absorption on the surface of cell through oxidation by LiP enzymes and by accumulation inside biomass of living cells, with practical future applications, is achieved.

## 2. Materials and Methods

### 2.1. Sampling and Isolation Process of Functional Bacteria

Soil samples were collected from the plant root area of Kyonggi University. After sieving, 5 g of soil was added to 50 mL of distilled water. One milliliter of the mixed solution was added to a medium that contained 5 g/L glucose and 5 g/L yeast extract as the main carbon and nitrogen sources for bacterial growth, respectively [23,24], supplemented with basic salt components including 2 g/L $KH_2PO_4$, 0.05 g/L $MgSO_4 \cdot 7H_2O$, 0.01 g/L $CaCl_2 \cdot 2H_2O$, 0.01 g/L $CaCl_2 \cdot 2H_2O$, 0.01 g/L $CuSO_4 \cdot 5H_2O$, and 2 g/L $Na_2HPO_4$ (pH 7–7.5). The culture samples were incubated at 30 °C for 2 weeks at 120 rpm in a shaking incubator. Subsequently, 100 μL was spread on agar plates that contained the same nutrient

components as the previous liquid medium. Each pure colony was separated into a new agar plate and was used as a material for the next analysis.

### 2.2. Screening of MB-Detoxifying Bacteria

Each pure colony was screened for Lignin peroxidase (LiP) on an agar plate containing 15 g/L agar, 5 g/L yeast extract, 5 g/L glucose, 2 g/L $KH_2PO_4$, 0.05 g/L $MgSO_4 \cdot 7H_2O$, 0.01 g/L $CaCl_2 \cdot 2H_2O$, 0.01 g/L $CaCl_2 \cdot 2H_2O$, 0.01 g/L $CuSO_4 \cdot 5H_2O$, and 2 $Na_2HPO_4$ (pH 7–7.5). MB was added at a final concentration of 1000 mg/L when the agar cooled to 60 °C after autoclaving at 121 °C for 20 min. All of the samples were incubated at 30 °C for one week. Isolates that produced clear decolorization zones were selected for further study.

### 2.3. Characterization and Identification of Isolated Bacteria

Molecular identification was performed using 16S rRNA gene sequencing. The 27F and 1492R universal primers were used to amplify the 16S rRNA gene, and sequencing was performed with Macrogen using 785F and 907R primers [25]. The PCR products were purified using a multiscreen filter plate (Millipore Corp., Bedford, MA, USA) and were then sequenced using 518F (50-CCAGCAGCCGCGGTAATACG-30) and 800R (50-TACCAGGGTATCTAATCC-30) primers with the PRISM BigDye Terminator v3.1 Cycle Sequencing Kit (Applied Biosystems, Foster City, CA, USA). This process was carried out at 95 °C for 5 min. The product was analyzed using an ABI Prism 3730XL DNA analyzer (Applied Biosystems, Foster City, CA, USA) after cooling on ice for 5 min. The full-length 16S rRNA sequence was assembled using SeqMan software (DNASTAR Inc., Madison, WI, USA). The related sequences were compared to this sequence using data from the EzBioCloud database (http://ezbiocloud.net, accessed on 7 July 2021) [26].

The related 16S rRNA sequences with the strain *Bacillus* sp. React3 were obtained from the GenBank database in order to construct a phylogenetic tree. The sequences were aligned, and phylogenetic trees were reconstructed using the MEGA 7 program with the output alignment file obtained from ClustalW [27]. The Tamura two-parameter model with gamma-distributed rates plus invariant sites based on the minimum Bayesian information criterion value of 140 (gamma parameter = 0.6) was determined as the best-fit model in this study for the neighbor-joining (NJ) analysis [28]. The reliability of the phylogenetic trees was estimated using bootstrap values of 1500 replications [29]. The obtained 16S rRNA sequence of the *Bacillus* sp. React3 was registered to the GenBank database with accession number OL440405.

### 2.4. Growth Curve of the Bacillus sp. React3 Isolate

The bacterial population of the *Bacillus* sp. React3 was calibrated for 20 h (1 h intervals) with the number of viable bacteria plotted in a graph, which was illustrated by a logarithmic growth curve. A prepared lignolytic bacterial culture of $10^4$ cell/mL according to the dilution factor, was added in 100 mL of sterilized lignin broth and was incubated at 35 °C for 20 h with continuous shaking at 120 rpm. The culture suspension followed by serial dilution was measured at a wavelength of 600 nm to determine the CFU/mL, at intervals of 1 h of incubation.

### 2.5. Optimization of Lignin Peroxidase Enzyme Activity

The effect of the variable factors on LiP production was evaluated under different incubation times over 48 h (6 h intervals), pH of 4–9 (1 unit intervals), substrate concentrations from 500 mg/L to 2000 mg/L (500 mg/L interval), and temperatures from 25 °C to 40 °C (5 °C interval).

### 2.6. Removal Test
2.6.1. Effect of the Temperature and pH

Decolorization of MB was studied at 25 °C, 30 °C, 35 °C, and 40 °C for 48 h at pH ranging from 4 to 9 (1 unit intervals) adjusted with HCl or NaOH 0.1 N. To investigate

the optimal pH value for the discoloration of MB, experiments were conducted under the optimal MB concentration and temperature. The inoculum used in this pH study was $1 \times 10^6$ CFU/mL. The absorbance of the samples was measured spectrophotometrically at 660 nm.

### 2.6.2. Effect of the Methylene Blue Concentration

MB (Samchun Pure Chemical Co., Ltd., Seoul, Korea) was prepared at serial concentrations of 200, 500, 1000, 1500, and 2000 mg/L for the removal test. The medium culture containing MB was inoculated with 4% concentration of the *Bacillus* sp. React3.

### 2.6.3. Effect of the Inoculum Concentration

To optimize the decolorization percentage, different inoculum concentrations were set, ranging from 1–5% ($v/v$). The samples were then incubated at 35 °C for 48 h.

### 2.6.4. Effect of Agitation

Experiments were performed under two conditions, static and shaking, at 100, 120, and 150 rpm at 35 °C for 3 days.

### 2.6.5. Effect of the Carbon and Nitrogen Sources

To enhance the MB dye decolorization by strain React3, different carbon and nitrogen sources were used at 1% concentration in the modified MB medium, including 0.1% MB ($w/v$), 2 g/L $KH_2PO_4$, 0.05 g/L $MgSO_4{\cdot}7H_2O$, 0.01 g/L $CaCl_2{\cdot}2H_2O$, 0.01 g/L $CaCl_2{\cdot}2H_2O$, 0.01 g/L $CuSO_4{\cdot}5H_2O$, and 2 g/L $Na_2HPO_4$ (pH 7–7.5). The samples were incubated at 35 °C for 3 days. The carbon sources used in this study included glucose, lactose, starch, tryptone, and maltose. The nitrogen sources included yeast extract, sodium nitrate, peptone, and urea.

### 2.7. Enzymes Activity Assays

After centrifuging at $8000 \times g$ and 4 °C for 10 min, the bacterial culture supernatant was used to determine the enzyme activity. The lignin peroxidase (LiP) activity was performed with the final reaction mixture (4 mL) including 1.5 mL ($v/v$) sodium tartrate buffer (125 mM, pH 3.0), 0.5 mL ($v/v$) MB dye (0.160 mM), 1.5 mL supernatant of bacterial culture, and 0.5 mL ($v/v$) of $H_2O_2$ (2 mM) in a reaction cuvette. The reaction mixture was incubated at 25 °C for 20 min after initiating by adding $H_2O_2$. The oxidation of the MB dye was observed at 310 nm using a spectrophotometer. Tubes without inoculum were run as the controls [30].

One unit of enzyme activity was defined as the amount of active enzyme required to convert 1 μmol of substrate into product per minute.

### 2.8. MB Decolorization Assay

Aliquots were collected after 1–6 h (1 h intervals) from flasks and were centrifuged at $8000 \times g$ and 4 °C for 10 min. MB degradation was determined by measuring the absorbance of the bacterial culture supernatants at 665 nm using a UV–Vis spectrophotometer. For comparison, the control without inoculum was prepared in the same culture medium. The experiments were performed in triplicate, and discoloration efficiency was calculated using the following equation [31]:

$$\text{MB degradation } (\%) = \frac{\text{Initial absorbance} - \text{Absorbance after degradation}}{\text{Initial absorbance}} \times 100 \quad (1)$$

The experiments for the removal tests and enzyme production were performed in triplicate, and the average was considered.

## 3. Results

### 3.1. Characteristic and Phylogenetic Identification of Bacterium

Morphologically, *Bacillus* sp. React3 was Gram-positive, rod-shaped, facultative anaerobic, irregular, with white colonies, and it grew satisfactorily in a medium containing 10 g/L tryptone, 5 g/L yeast extract, and 10 g/L sodium chloride. The 16S rRNA gene sequencing analysis revealed that this strain is a member of the genus *Bacillus* and is closest to *Bacillus velezensis* CR-502$^T$, with a similarity of 98.86% [32] (Figure 1).

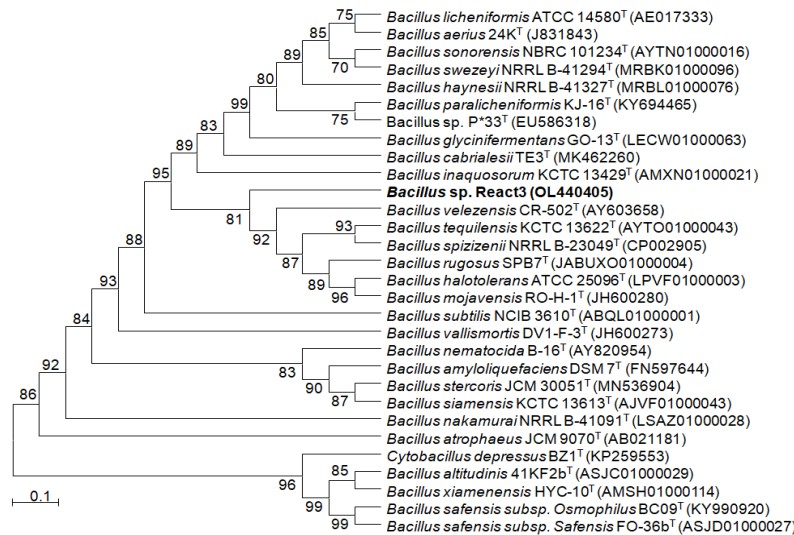

**Figure 1.** Phylogenetic tree of *B.* React3 and related members in the genus *Bacillus*.

### 3.2. Potential LiP Enzyme Production

*Bacillus* sp. React3 was found to be capable of producing a clear decolorization zone around colonies on the agar containing dyes from day 2 at 35 °C. The presence of a maximum white decolorization zone on the agar plates supplemented with MB on day 5 indicated an LiP enzyme activity (Figure 2).

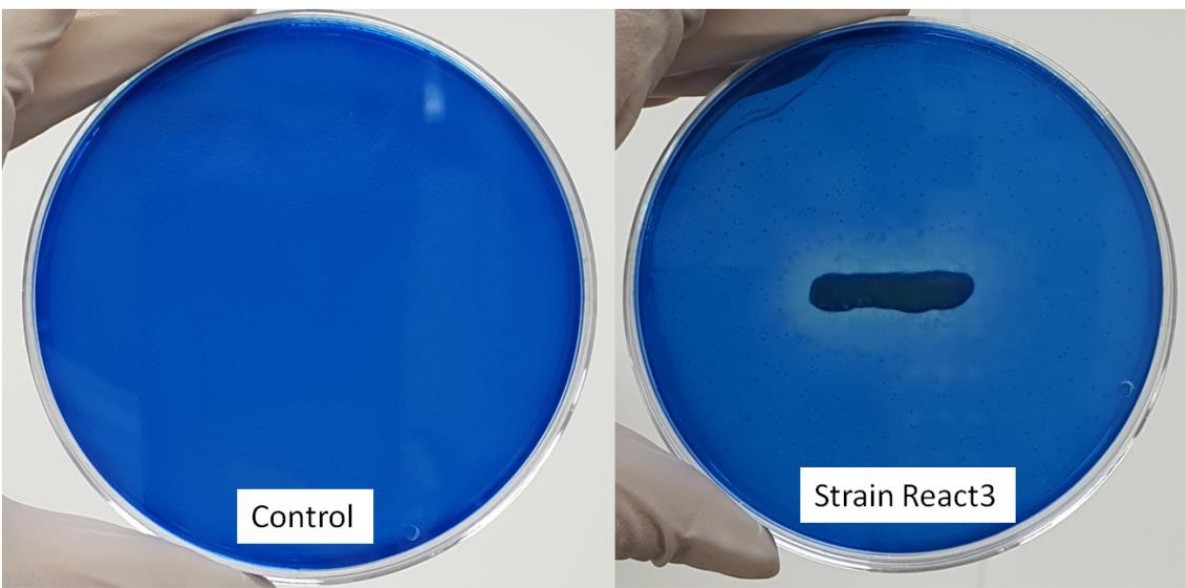

**Figure 2.** Qualitative screening of lignin peroxidase enzyme production on agar plates containing 1000 mg/L of methylene blue.

### 3.3. Optimal Conditions for Growth and Enzyme Production of B. React3

The action of the produced enzyme may be responsible for the effective decolorization of MB dye within 6–48 h. The enzyme activity increased with increasing the incubation time and bacterial growth (Figures 3 and 4). The maximum LiP production was recorded during decolorization after 36 h of static incubation with 45 U/mL, while it was observed to be 44 U/mL after 18 h under shaking incubation at 120 rpm, at 35 °C and pH 7.

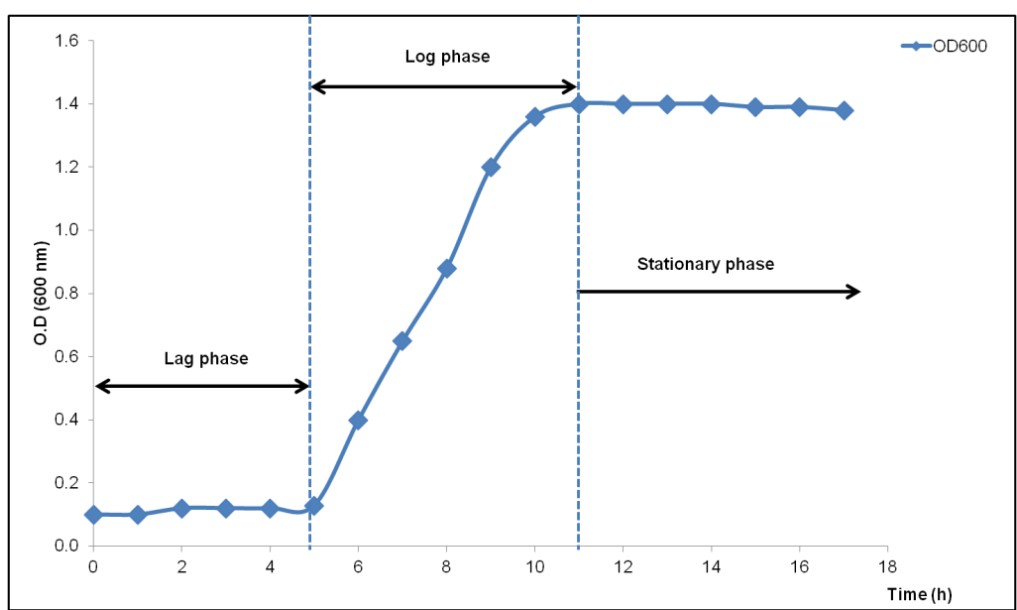

**Figure 3.** Bacterial growth curve of *Bacillus* sp. React3 after 20 h of incubation at 120 rpm and 35 °C.

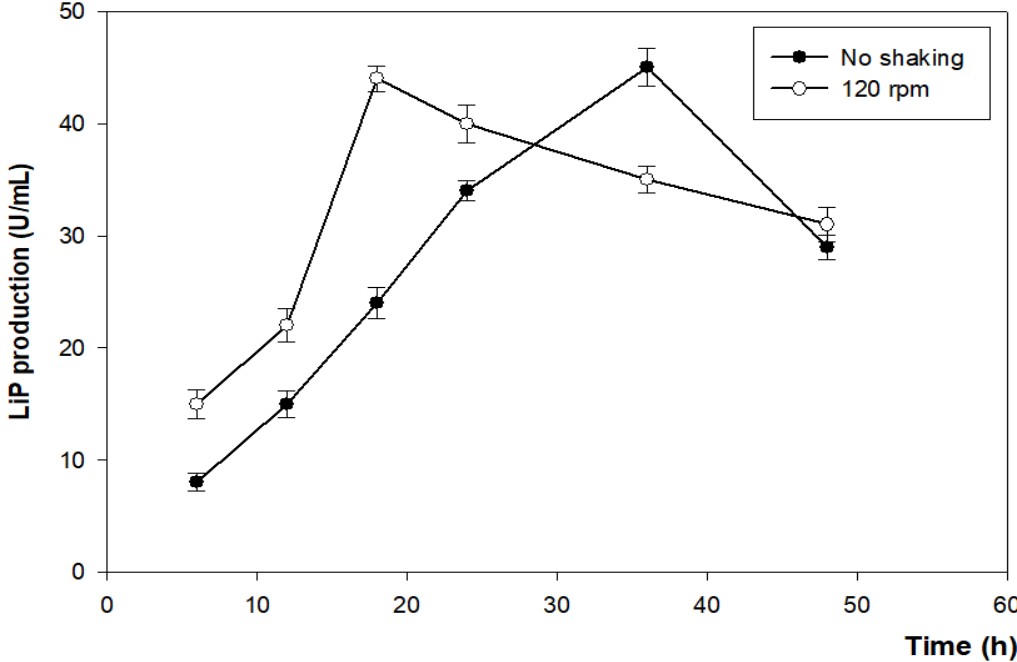

**Figure 4.** LiP activity during the decolorization of dyes after 48 h at 35 °C.

### 3.4. Effect of Different Factors on Decolorization of MB Dye

3.4.1. Effect of Temperature and pH

The decolorization activity was determined by spectrophotometric measurement of the absorbance of the bacterial culture crude extract at a wavelength of 665 nm for MB

absorption [33]. The absorbance of the culture supernatant of *Bacillus* sp. React3 compared with broth without inoculation as a control demonstrated a similar reduction of the peak at a wavelength of 665 nm, with a decolorization percentage of 99.5% (Figure 5).

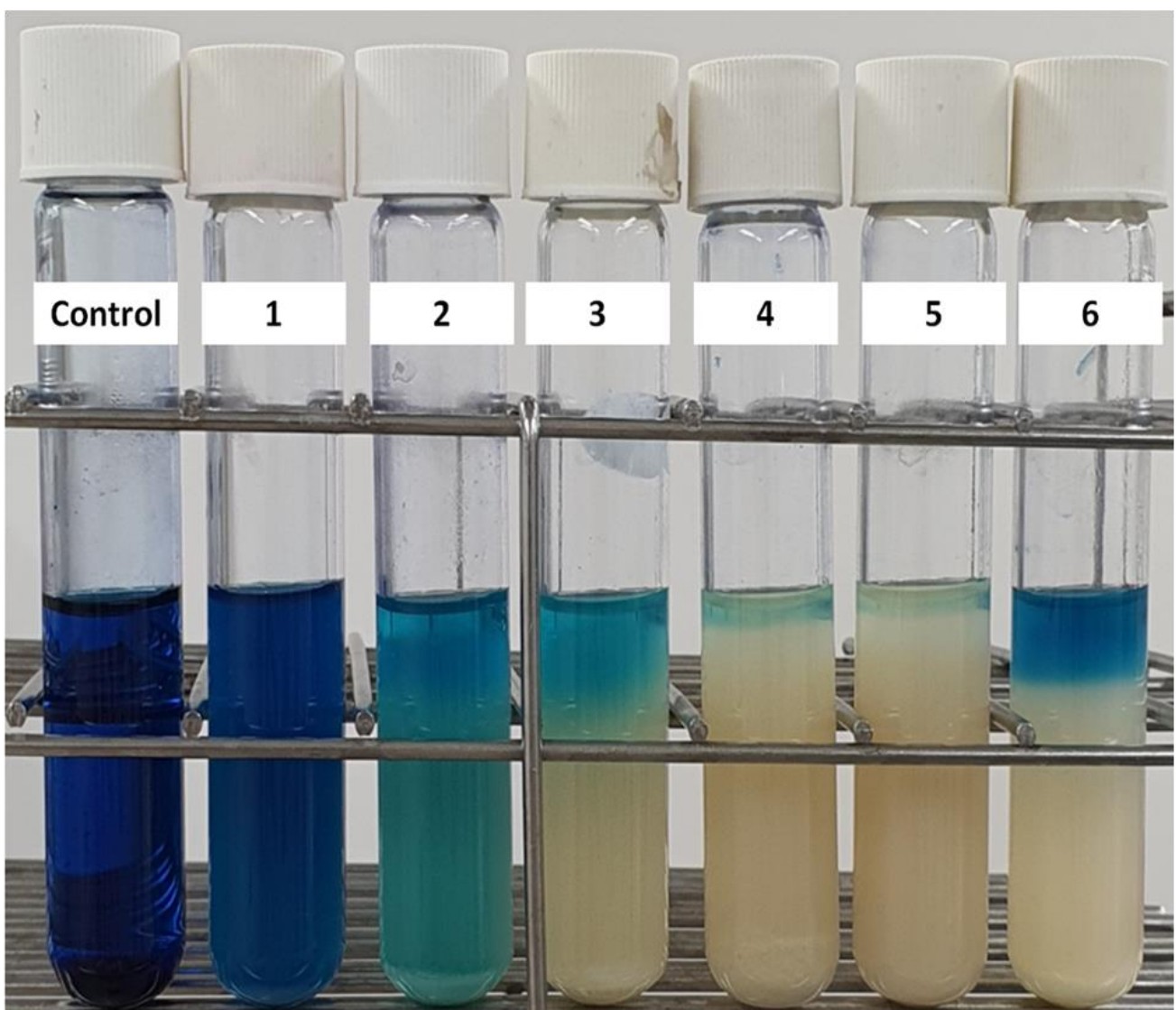

**Figure 5.** Decolorization of MB by *Bacillus* sp. React3 after 48 h at 35 °C. Samples: (control) no inoculums; (1) 1% bacterial inoculum incubated at 120 rpm; (2–6), 1−5% bacterial inoculum incubated at a static condition.

We observed that the decolorization efficiency of *Bacillus* sp. React3 gradually increased with the increase in temperature, peaked at a maximum removal of 100% at 35 °C from 6 h to 12 h of incubation, but began decreasing once the temperature was over 35 °C, as shown in Figure 6a. The maximum decolorization was obtained at pH 7, whereas its activity was slow at either an acidic or alkaline pH (Figure 6b). The decolorization efficiencies determined with 65% and 82% were at pH 5 and 6, respectively. At pH 8, the efficiencies were determined to be higher at 89%, but decreased to 65% at pH 9.

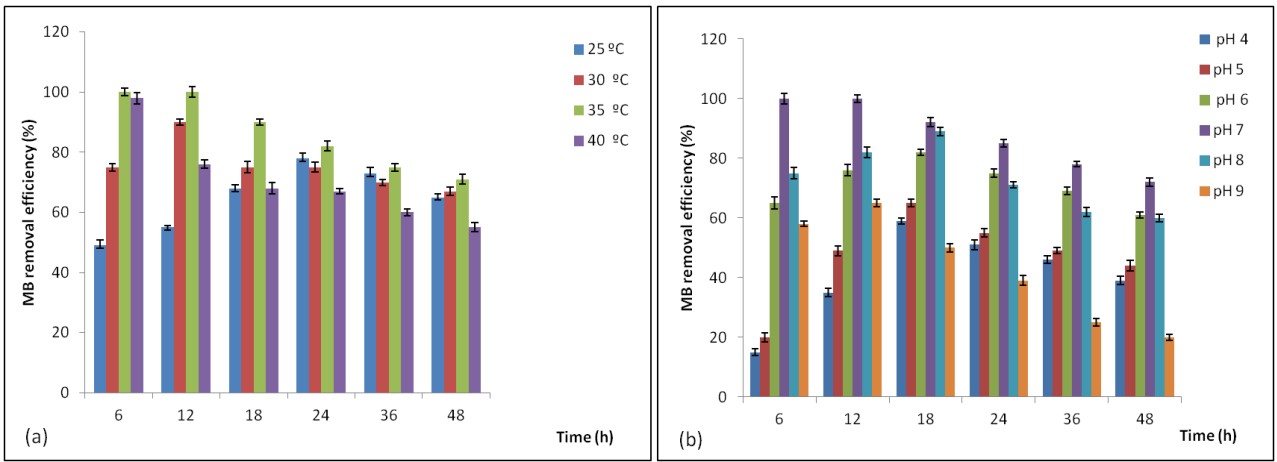

**Figure 6.** The effect of temperature (**a**) and pH (**b**) on the decolorization of MB dye.

### 3.4.2. Effect of Initial MB Concentration

The recorded decolorization rate of MB varied with different initial MB concentrations, as shown in Figure 7. Decolorization obtained the maximum degree of 100% with an initial dye concentration from 200 to 1000 mg/L. A further increase in dye concentration resulted in a reduction in the decolorization rates, with 66% at 1500 mg/L and 50% at 2000 mg/L after 48 h.

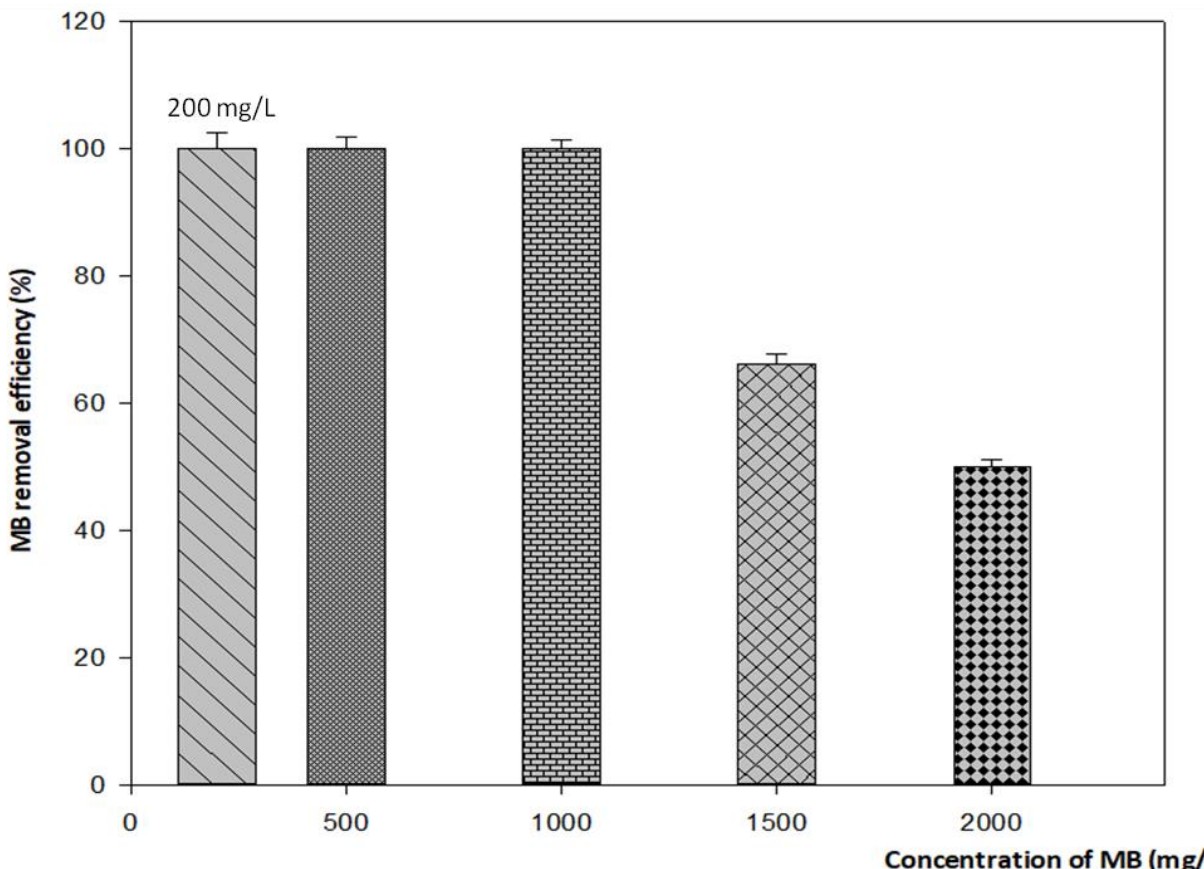

**Figure 7.** MB removal efficiency at different concentrations of initial MB with 4% inoculum incubated at 35 °C for 48 h.

### 3.4.3. Effect of Inoculum Size

The decolorization rate increased when increasing the inoculum size, reaching a maximum rate of 99.5% at 4% (*w/v*) inoculum and 1000 mg/L MB under static conditions after 48 h (Figure 8). However, the inoculum size and removal efficiency did not vary considerably.

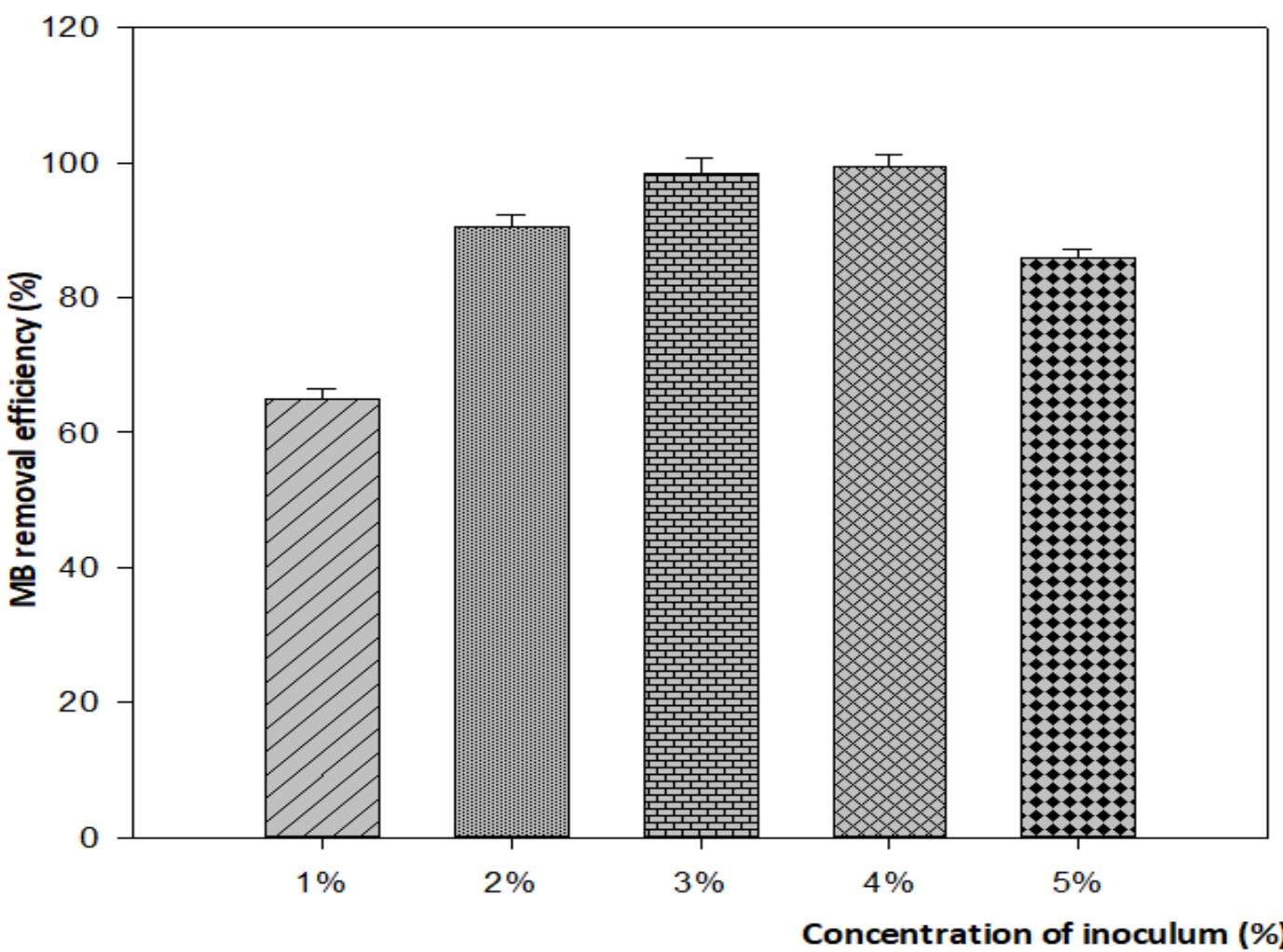

**Figure 8.** MB removal efficiency at various concentration of inoculums, with an initial 1000 mg/L MB incubated at 35 °C for 48 h.

### 3.4.4. Effect of Agitation

It is clear that the decolorization of all of the dyes tested was higher than that under agitation in the initial period of bacterial growth of 6 h at 35 °C. After shaking at 120 rpm, a decolorization of 100% was recorded, whereas it achieved only 35% under static conditions within the same incubation time, as shown in Figure 9a,c. The performance of the 1000 mg/L = MB removal was 85% with a shaking speed of 100 rpm, and 78% with a decreased speed of 150 rpm. However, the MB degradation of the shaking condition decreased after measurement at 12 h of incubation (Figure 9b).

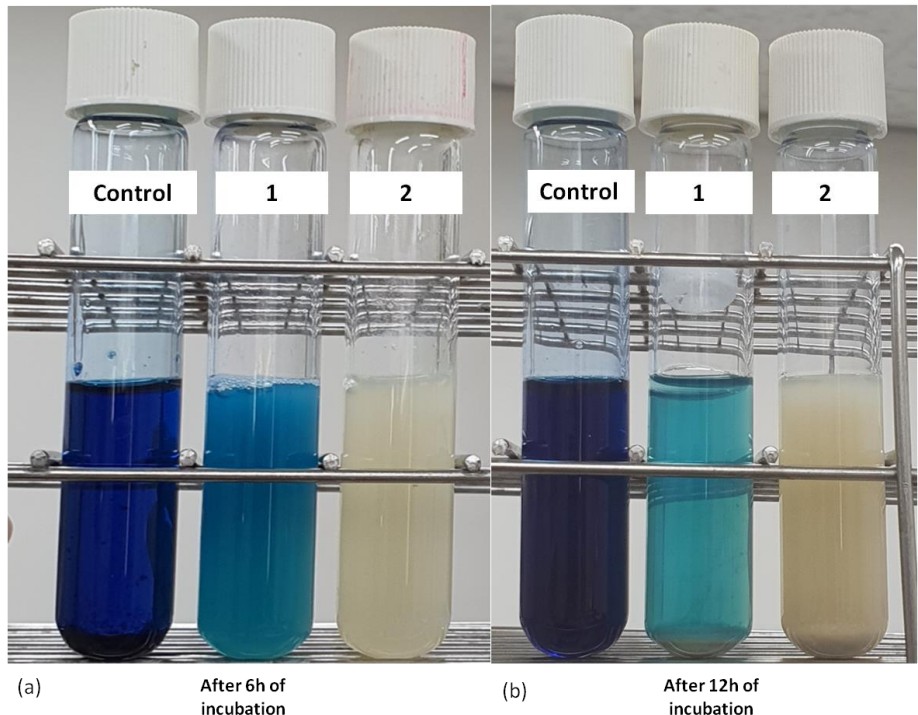

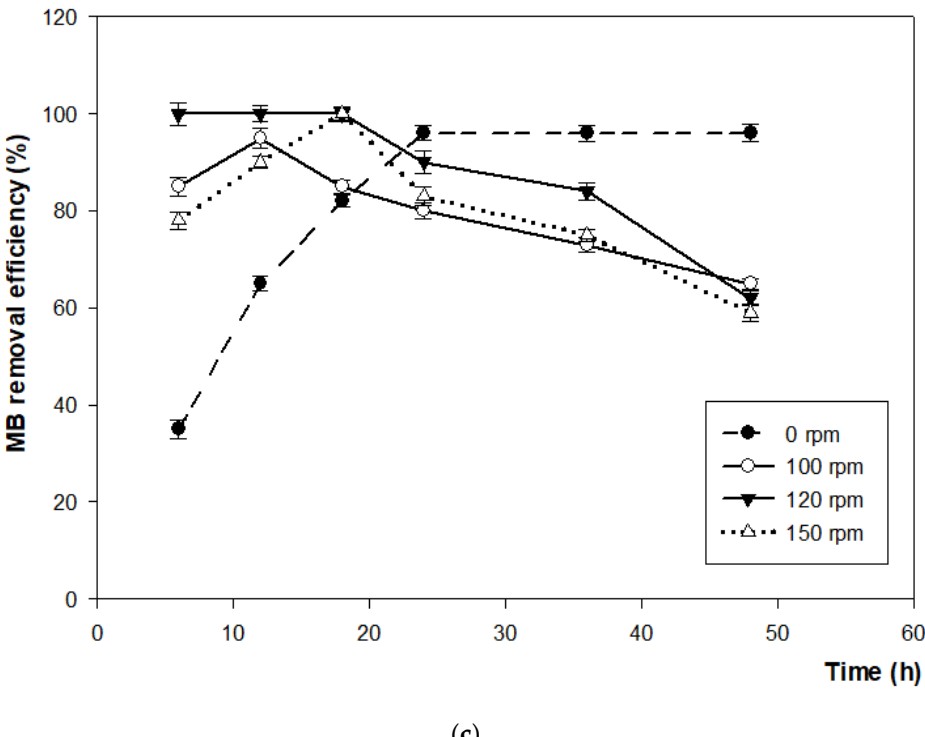

**Figure 9.** MB removal efficiency using 4% inoculum at 35 °C. After 6 h (**a**); 12 h (**b**): (Control), no inoculum; (1) incubated at static and (2) at 120 rpm, and (**c**) at static and different speeds of shaking after incubation for 48 h.

### 3.4.5. Effect of Different Carbon and Nitrogen Sources

Figure 10 shows the decolorization of the 1000 mg/L MB concentration by React3 using various carbon and nitrogen sources recorded after 12 h of incubation at 35 °C for 48 h. The maximum decolorization (95%) of MB dye was observed in the presence of tryptone,

followed by glucose (83%), starch (75%), lactose (67%), and maltose (60%) (Figure 10a). Yeast extract was found to be the optimal nitrogen source for MB removal, which was observed to be 97%, followed by peptone (80%), sodium nitrate (66%), and urea (60%) (Figure 10b).

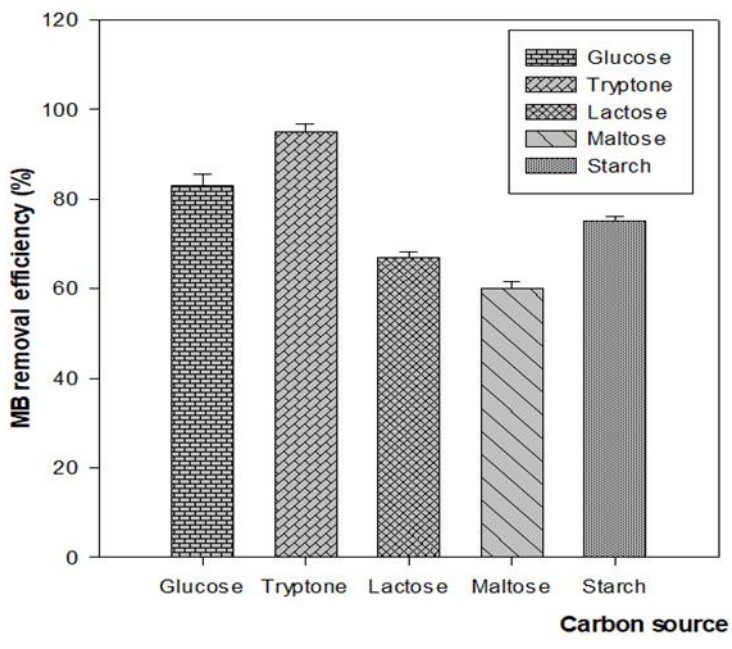

(**a**)

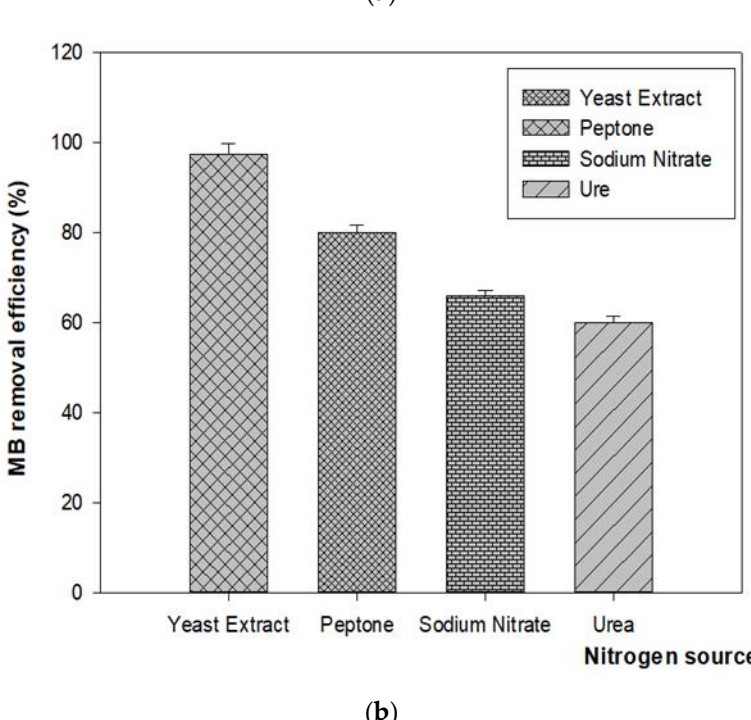

(**b**)

**Figure 10.** Different carbon (**a**) and nitrogen (**b**) sources regarding MB decolorization efficiency using *Bacillus* sp. React3 at 35 °C for 48 h.

## 4. Discussion

Several *Bacillus* strains have been reported to degrade lignin, as evidenced by the potential production of ligninolytic enzymes, such as laccase, from *B. pumilus* C6 and *B. atrophaeus* B7 [34]. *Bacillus* sp. LD003 has been sustained on lignin fractions and was able

to decolorize various dyes, including Azure B, MB, and toluidine blue O [35]. The closest strain, *Bacillus velezensis* CR-502[T], to strain React3, was reported as a member belonging to the operational group *Bacillus amyloliquefaciens*, which is known as a plant growth-promoting bacterium that produces various enzymes, including α-amylase, protease, lipase, cellulase, xylanase, pectinase, aminotransferase, barnase, peroxidase, and laccase [36]. However, its dye degradation ability has not yet been investigated. Therefore, this study first found the potential novel bacterial strain named React3, which is capable of MB removal with a high efficiency of 99.5% after 48 h through LiP in the oxidation process of MB.

The decolorization of MB was at a maximum at 100–120 rpm, which may be due to the proper mixing of nutrients and oxygen in the medium in which the bacterial strain accelerated its growth and enzyme production for degrading MB. Roy et al. isolated an *Enterobacter* sp. CV–S1 from textile industry effluents involved in the effect of crystal violet dye under aerobic shaking conditions at 120 rpm [37]. In another study, a removal rate of MB dye of 99.27% with a 0.1% concentration was achieved by shaking at 100 rpm with *Bacillus albus* MW407057 [3]. Similar results were also reported for *Aeromonas hydrophila* for the removal of crystal violet dye [38]. Interestingly, this study illustrated that MB was removed efficiently, and reached 100% after 6 h, because at the point of the log phase, bacterial growth starts under shaking incubation. Meanwhile, MB was degraded completely after 24 h under non-shaking conditions. However, the removal efficiency of MB decreased under shaking conditions at 120 rpm, while it almost remained stable during the lag phase of bacterial growth without shaking conditions. It is suggested that the mechanism of MB bioremediation of the bacterial strain React3 is similar for living cells as that of the accumulation. The accumulation process of MB in the living cells and via the oxidation process by LiP is more efficient and stable under static than shaking conditions over time, especially in the dead phase of bacterial growth with more dead cells (Figures 5 and 9a,b) [38].

The optimal temperature for MB degradation in our study was 30 °C, which is in the optimal range of 30–40 °C for dye degradation in other studies [39]. In a previous study, the complete removal of crystal violet was obtained at 35 °C; however, it decreased to 37.5% at 30 °C and 40 °C [38]. Holey [40] reported that bacterial consortium exhibited 98% decolorization of Congo red at 37 °C. However, the optimal temperature for malachite green degradation by *Bacillus cereus* KM201428 was observed at 40 °C [41]. High temperatures distort the structure of the enzyme, thus diminishing the binding capacity of its active sites to the substrate molecules [42]. Moreover, this might be due to a decrease in microbial growth and enzyme activity at low temperatures, causing a low decolorization efficiency [43].

Several researchers have shown that biosorption processes using microbes are highly pH-dependent [44,45]. Moreover, isolates with a wide range of pH tolerance conferred advantages over isolates with a narrow pH tolerance. The results obtained in this study were the same as those of other studies, the degradation rate of the dye increased with the increasing pH, reaching a maximum point at pH 6.0–7.0; subsequently, the efficiency decreased when the pH value increased [46]. Crystal violet dye was effectively removed at pH 7 by a ligninolytic enzyme producing *Aeromonas hydrophila* [47]. In these cases, higher pH values may have caused the degradation rate to decrease, owing to the effect of the enzyme activities and bioactivity [48]. However, in other studies, a range of pH from to 9–10 was determined to be the optimal for dye decolorization using *Bacillus* sp. strain CH12 and *Exiguobacterium profundum* strain CMR2 [48,49]. However, under an acidic pH, the H[+] ions may inhibit the dye cations, causing a reduction in the decolorization performance [3].

This clearly indicates that the efficiency of decolorization greatly varies according to different carbon and nitrogen sources, which strongly depends on the demand of various bacterial strains. Starch was the optimal carbon source for bacterial strain *Exiguobacterium profundum* strain CMR2 to maximize the removal efficiency of Reactive Blue EFAF, which reached 82% [49]. Nevertheless, the maximum decolorization of MB dye was 98.11% in the presence

of glucose by *Bacillus albus* MW407057 [42]. In the case of nitrogen sources, a similar result for the maximum rate of MB dye degradation was determined for the contribution of yeast extract in a previous study [42].

High concentrations of MB may be attributed to its toxicity to bacterial cells, inhibition of metabolic activity, and saturation of the cells with dye products. Additionally, there might be inactivation of the transport system by the dye or blockage of the active site of azoreductase enzymes by the dye molecules [50].

The catalytic enzymes produced by microbes are considered promising bio-agents that can catalyze pollutant degradation [44]. In the present work, LiP from the *Bacillus* sp. React3 was investigated during a screening test for MB removal. As a ligninolytic enzyme, LiP can degrade various synthetic dyes, phthalates, and polycyclic aromatic hydrocarbons [46,51]. Azure B dye decolorization was determined to be as high as 90% by ligninolytic strain *Serratia liquefaciens* [51], and crystal violet was decolorized by LiP extracted from *A. hydroohila*.

## 5. Conclusions

A LiP-producing bacterium, *Baccilus* sp. React3, was isolated from the soil and identified as *Bacillus* sp., with accession number OL440405. This bacterial strain was able to degrade 1000 mg/L concentration of MB dye up to 100% until 6 h, and maintained stability until 12 h before decreasing under a shaking condition at 120 rpm, and reached 99.5% within 48 h without shaking at optimal conditions with 4% inoculum at 35 °C and pH 7, with tryptone as the carbon source and yeast extract as the nitrogen source. Additionally, this strain could grow in a wide range of pH (5–10), temperature (5–55 °C), and NaCl (0–25%) conditions, such that various potential applications in different fields could be predicted, not only for dye decolorization in wastewater treatment. Interestingly, *Bacillus* sp. React3 has shown the novel mechanism of MB removal through both absorption on the surface of bacterial cells and through accumulation inside the cells under different agitations. However, further studies, including purification, identification of enzymes, and cytotoxicity assays, should be conducted in the future to discover more about the potential of this bacterial strain before its application on a large scale.

**Author Contributions:** Conceptualization, V.H.T.P.; methodology, V.H.T.P.; validation, V.H.T.P.; formal analysis and investigation, V.H.T.P.; resources, W.C.; writing—original draft preparation, V.H.T.P.; writing—review and editing, V.H.T.P. and J.K.; supervision, W.C. and S.C.; funding acquisition, S.C. and W.C. All authors have read and agreed to the published version of the manuscript.

**Funding:** This work was carried out with the support of 'Cooperative Research Program for Agriculture Science and Technology Devel opment (Project No. PJ01529302)' Rural Development Administration, Korea.

**Institutional Review Board Statement:** Not applicable.

**Informed Consent Statement:** Not applicable.

**Data Availability Statement:** The data used to support the findings of this study are available from the corresponding author upon request.

**Conflicts of Interest:** The authors declare no conflict of interest.

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
