# Peer review of "Biodegradation of Methylene Blue Using a Novel Lignin Peroxidase Enzyme Producing Bacteria, Named Bacillus sp. React3, as a Promising Candidate for Dye-Contaminated Wastewater Treatment"

_fermentation, doi:10.3390/fermentation8050190_

Round 1

Reviewer 1 Report

The manuscript presented a Biodegradation of Methylene Blue using a novel lignin peroxidase enzyme producing bacteria, named Bacillus sp. React3, promising candidate for dye-contaminated wastewater treatment. The paper has been written quite clearly and is easy for understanding. I suggest the following corrections:

  1. Abstract and Introduction: Was very well written.
  2. Materials and Methods: Was well written, you have to pay attention to some grammatical errors in English.

Page 3, line 125. Please always report the denomination Bacillus sp. React3. Change it also in the other parts of the paper.

Page 4, lines 162,173. Please convert the rpm value to g.

  1. Results: Please revise the English and write each part of the results section in more detail. Have you thought about doing a statistical analysis of the data?

Page 5, line 191. Have you tried measuring the diameter of the clarification halo to get semi-quantitative values on a plate, at the screening phase?

In general, is it possible to get some data on the Screening of MB-detoxifying bacteria isolated from the soil?

Page 7, Figure 6. Please review the figure and in particular please add the symbol °C next to the temperatures.

Page 8, Figure 7. Please review the figure and in particular please change the “concentration of the inoculum (%)” with “concentrations of initial MB (%)”.

Page 8, Figure 8. Please review the figure.

Page 9. Figure 9. Please review the figure. Please revise the figure to make it clearer to understand.

Page 10. Figure 10. Please review the figure.

  1. Discussion: The discussion is well written, you have to pay attention to some grammatical errors in English.
  2. Conclusion: Should be more effective and should stress the importance and novelty impact also for the possible applications on large scale.

Author Response

Dear Reviewer,

We sincerely thank you very much for your time to review our manuscript with valuable comments. We have revised all things as you recommended in the yellow highlight. Please kindly to find our response letter in the attachment.

All the best,

Van Pham

Reviewer 2 Report

Paper title “Biodegradation of Methylene Blue using a novel lignin peroxidase enzyme-producing bacteria, named Bacillus sp. React3, a promising candidate for dye-contaminated wastewater treatment.” By Pham et al. describe the efficacy of Bacillus sp. isolated from the soil sample in the degradation of MB. The manuscript contains promising data, but I have some major issues that should be addressed before consideration for publication in the Fermentation journal.  

  • The abstract should be rephrased, Line 14 – 20 should be summarized in one statement to find a space to enumerate your results. the right place of accession number is the material and method section, subsection 2.3 not in the abstract.
  • Line 28: the scientific name “Bacillus” should be italic.
  • Keywords should be rephrased to contain highlight words such as Biodegradation; MB, lignin peroxidase enzymes; Bacillus sp.; wastewater.
  • Line 32: delete “Various”.
  • Line 35: “dye, called” please delete “, called”
  • Line 37: correct the symbol “(SO−3)”
  • Line 57: “dye” word is repeated, please delete the second one.
  • The main hypothesis of the study should be rephrased at the end of the introduction section.
  • It is known to isolate high degrading bacterial strains, it is logical to collect contaminated soil with dyes or other contaminants, but collecting soil samples from root plants, will isolate bacterial strains with low capacity to dye degradation, please clarify the reason for collecting samples from root plant.
  • Line 92: authors said “Subsequently, 100 mL of it were spread”, who 100 mL of liquid spread over agar plate, it is too large, please check?
  • Subsection 2.2., please specify the concentration of MB.
  • Due to the absence of a species name, the B. React3 should be written completely as “Bacillus React” not “B. React3”.
  • Line 128: “104 cell/mL” it should be “104 cell/mL” please check.
  • It is better to adjust the pH using buffers because bacterial metabolites can return the medium to neutral when adjusted by HCL or NaOH, please take this point into your consideration in the future.
  • Line 135: what is the meaning of substrate concentration, if it is MB, the author in lines 145 and 146 mentioned that the MB concentration was 200, 500, …..2000 mg/L, please clarify.
  • Line 183: “gram-positive” should be “Gram-positive”.
  • Line 186: the scientific name “Bacillus” and “Bacillus velezensis” must be italic.
  • Line 192: “after 2-5 days”, then the clear zone observed after 2, 3, 4, or 5 days? Please specify.
  • The title of Figure 2 should be, Qualitative screening…..
  • Theoretically, the bacterial growth curve should be containing lag, log, stationary, and death phase, but practically, how the O.D value decreased especially the spectrophotometer read the O.D. for live and dead cells. therefore, the O.D. should not decrease with time because the cells are present either alive or dead, please clarify this phenomenon in Figure 2 (please correct me if this is wrong)
  • Line 212: “B.” please write it complete and in italic.
  • Please check the title of Figure 5, something is wrong.
  • Please add error bars in Figures 6a and b.
  • Figure 7 is wrong, the present figure for inoculum size not for MB concentration, therefore it is identical to figure 8. Please revised.
  • Line 258: The concentration of MB in the carbon and nitrogen experiment was 1% or 200 ppm as used in the inoculum size experiment. Please clarify.
  • The manuscript contains several typo-error and grammatical mistakes, please revised the manuscript carefully.

Author Response

(The authors gave the same response as above.)

Round 2

Reviewer 2 Report

Thanks to the editor and the Authors. The authors have addressed major raised comments. It is my pleasure to recommend this article for publication after minor revision.
- The Label of the first column in Figure 7 (200 mg/L) should be added.
Please check the scientific names in the reference section, they should be italic, such as references 9, 41 (bacillus should be Bacillus), 44, and 45

Author Response

Dear Reviewer,

Once again, we would like thank you very much for your review with valuable recommendations on our manuscript to make it acceptable for the publication.

  • We have added 200 mg/L as the label in the first column of Figure 7.
  • We have corrected all names of microorganism  in Italic in the references 9, 41  44, and 45.

The best wishes,

Van Pham